# Compliance of Adolescent Friendly Health Clinics with National and International Standards: Quantitative findings from the i-Saathiya study

Deepika Bahl [ID],[1] Shalini Bassi [ID],[1] Heeya Maity,[1] Supriya Krishnan,[1] Stefanie Dringus,[2] Amanda Mason-Jones [ID],[3] Anku Malik,[1] Monika Arora [ID][1]

DB and SB contributed equally.

¹Health Promotion Division, Public Health Foundation of India, New Delhi, Delhi, India
²Independent Global Health Consultant, London, UK
³Health Sciences, University of York, York, UK

**Correspondence to**
Dr Deepika Bahl;
deepika.bahl@phfi.org

## ABSTRACT

**Objective** Indian adolescents experience several health challenges requiring acceptable, equitable, appropriate and effective healthcare services. Our objective was to assess the compliance of Adolescent Friendly Health Clinics (AFHCs) in two of India's largest states, using both national benchmarks (under Rashtriya Kishor Swasthya Karyakram-RKSK) and global standards (by WHO).

**Design** Cross-sectional study comprising structured observations and interactions (November 2021 to June 2022).

**Setting** Fourteen AFHCs across all levels of health system were included from two districts of Maharashtra (n=8) and Madhya Pradesh (n=6). These AFHCs were observed using checklist, and few items of checklist were verified by interactions with AFHC's health workers (medical officers/auxillary nurse midwives/counsellors) handlings adolescents. The developed checklist included 57 items based on adapted global standards and 25 items using national benchmarks.

**Result** High compliance of AFHCs with RKSK's benchmarks was attributed to various items including the accessibility through local transport (n=14, 100%), clean surroundings (n=11, 78.5%), presence of signage (n=10, 71.4%), convenient operating days and time (n=11, 78.5%), and secure storage of records (n=13, 92.9%). Concurrently, items that showed low compliance encompassed, the availability of Information, Education and communication (IEC) resources, which were deficient in 57.1% of AFHCs (n=8). Similarly, designated areas for clinical services (n=10, 71.4%) and commodity disbursement (n=9, 64.3%) lacked in more than half of the recruited AFHCs. Additionally, lack of guidelines for referrals (n=13, 92.9%), as well as standard operating procedures to ensure equity, non-judgemental attitude, competence, confidentiality and referral as per WHO standards.

**Conclusion** Evidence spotlights the strengths and gaps in AFHCs, aligning with, government's priorities on adolescent health. Addressing the identified gaps is crucial to creating healthcare facilities that are adolescent-friendly, easily accessible and effectively navigate adolescent health challenges. This concerted effort would contribute to their development and transformation, playing a pivotal role in India's progress.

## STRENGTHS AND LIMITATIONS OF THIS STUDY

⇒ The study comprehensively assessed the existing quality of adolescent-friendly healthcare services in India in compliance to both, national and global guidelines.
⇒ Since data were gathered only from two Indian states, the generalisability of the findings may be limited.
⇒ Integrating exit interviews with adolescent clients or employing mystery client methodology would have provided a comprehensive understanding of the healthcare worker's performance.

## INTRODUCTION

Adolescent Friendly Health Services (AFHSs) are evidence-based interventions to address health concerns by offering accessible, acceptable, equitable and appropriate healthcare services for adolescents.[1] In the South-East Asian Region, countries are prioritising adolescent health and well-being by scaling AFHS through their national public health systems. Multiple delivery settings and approaches including clinics, school health programmes, community outreach, partnerships with civil society, development partners and professional bodies are utilised to provide these services.[2] Evidence highlights that these adolescent-friendly services are cost-effective in delivering sexual and reproductive health (SRH) services to adolescents.[3]

In India, AFHSs are provided through Adolescent Friendly Health Clinics (AFHCs) as part of 'Rashtriya Kishor Swasthya Karyakram (RKSK)' (India's Adolescent Health Strategy) launched in 2014.[4] AFHC is one of the critical pillars of the RKSK that seeks to enable all adolescents to realise their full potential by making informed decisions concerning their health, and by accessing the services and support to implement their decisions. These clinics are meant to provide

a range of services listed as clinical and counselling services related to nutrition, injuries, non-communicable diseases, mental health, substance misuse and SRH to adolescents.[5] Integration of such services in the AFHCs is vital for accelerating the progress of adolescent health in India. The AFHCs have been integrated into medical colleges, district and subdistrict hospitals (SDH), community health centres (CHCs)/rural hospitals (RHs) and primary health centres (PHCs)[6] to provide comprehensive services for adolescents. Despite years of establishment, the available evidence from India highlights the low footfall of adolescents in AFHCs.[7–9] This is partly explained by multiple barriers to avail these services, including a lack of adolescents' knowledge about the AFHCs in their vicinity,[10] inadequate staffing, untrained service providers, social taboo, inadequate equipment, low priority accorded by programme managers and adolescents' perceive that services are restricted to reproductive and sexual health.[11] This results in a missed opportunity in the identification and treatment of health problems, hindering the assistance provided to adolescents in developing positive healthcare-seeking behaviours.[12] This undermines the goal of RKSK to promote adolescents' health and well-being and emphasises the pressing need for a standardised approach to enhance the quality of care across India,[11] ensuring effective utilisation by adolescents. With this, as part of the i-Saathiya study, which aims to explore the implementation of the peer education programme, we also assessed AFHC compliance with both national[13] and global standards for quality healthcare services catering to adolescents.[14] This assessment was carried out in the states of Madhya Pradesh and Maharashtra (India) through structured observations and interactions-based verifications. This method was suitable to achieve this objective as it allowed us to see all activities directly, rather than solely relying on subjective opinions and perceptions.[15] We envisage that the findings of this study will be useful in strengthening the Indian AFHCs, enabling them to safeguard, promote and improve adolescent health by providing acceptable, equitable, appropriate and effective healthcare services; and establishing safe and supportive environments to facilitate their growth and development.[16]

## METHODOLOGY

### Study setting and sample selection

AFHCs in two districts of Madhya Pradesh (Panna and Damoh) and Maharashtra (Yavatmal and Nashik) were observed. Panna and Damoh are situated in Madhya Pradesh, a large state in central India with a total population of 72.7 million and a literacy rate of 69.3%.[17] Yavatmal and Nashik are in Maharashtra, which is situated in the Western Peninsular region of India with an approximate population of 112.4 million and an 82.3% literacy rate.[17] The criteria for selection of states and districts have been described in another i-Saathiya publication.[18] Table 1 lists the AFHCs at various levels of the health system in

selected study districts and blocks of both states. To assess compliance, we included AFHCs from all the levels in Madhya Pradesh and Maharashtra. At the higher level, the number of AFHCs were limited, thus all were included for the assessment. One AFHC at the district-level (DH) or government medical college, in all four selected study districts (n=4) were included. Additionally each district of Maharashtra featured one AFHC at the SDH (n=2). But at the lower level of health facility, the number of AFHC was more than one. Thus, we adopted a random selection approach using a fishbowl method to choose one AFHC each from CHC or RH (n=4) and PHC from each study district (n=4). In total, 14 AFHCs were included in our study for assessment using this methodology.

### Study design

A cross-sectional study using quantitative methods was conducted between November 2021 and June 2022.

### Data collection

The trained study team conducted structured observations using a printed checklist with several performance criteria. This assessment was carried out in 14 clinics, out of 61 clinics in the selected districts to assess the compliance of AFHCs.

The checklist was developed by adapting the WHO's Global standards for quality healthcare services for adolescents to align with the Indian context[14] and using RKSK benchmarks.[13] The adaptation was carried through a Delphi process[19] involving panel of four health experts in adolescent health. Throughout the process, several items were excluded from the checklist, such as vaccine availability, ophthalmoscope, injectable contraceptives, refrigerator, contraceptive implants, etc. These exclusions were guided by the India's Adolescent Health Strategy (2014), which outlines the essential requisites for AFHCs.[6] The trained study team also completed the Good Clinical Practice course offered by National Drug Abuse Treatment Clinical Trials Network (NDAT CTN).

The developed checklist comprised a total of 82 items (WHO standards=57 items; RKSK Benchmarks=25 items). Some of the items such as standard operating procedures (SOPs), policy commitments, availability of condoms, contraceptives were observed and their responses were further confirmed through interaction with the concerned staff members (counsellors/auxillary nurse midwives (ANMs)/medical officers (MOs)) on the same day as per their convenient time without disturbing their day schedule. The embedded interactions approach was necessary, as specific items could be securely stored and concealed from direct observations. Online supplemental table S1 provides the list of items verified by the interaction with the above-mentioned staff due to their involvement in handling adolescents and their health issues within AFHCs. The printed checklist was pretested by the study team in one AFHC within each study state, to assess its feasibility in the Indian context. Feedback received from the pretesting primarily focused

**Table 1** Number* of AFHCs functional at various levels of health system in the study districts and blocks

| State | District | Blocks | Level of health facility | | | | |
| | | | Government medical college | District hospital | Sub district hospital† | CHCs/RH under each block | PHCs under each block |
| --- | --- | --- | --- | --- | --- | --- | --- |
| Madhya Pradesh | Panna | Panna | No | 1 | No | 1 | 4 |
| | | Ajaygarh | | | | 1 | 4 |
| | | Shahnagar | | | | 1 | 2 |
| | | Pawai | | | | 1 | 5 |
| | Damoh | Damoh | No | 1 | No | 1 | 2 |
| | | Jabera | | | | 1 | 2 |
| | | Patera | | | | 1 | 2 |
| | | Patharia | | | | 1 | 2 |
| Maharashtra | Yavatmal | Babulgaon | 1 | No | No | 1 | 2 |
| | | Pusad | | | 1 | No | 2 |
| | | Yavatmal | | | No | No | 2 |
| | | Zari-Jamani | | | No | 1‡ | 2 |
| | Nashik | Dindori | No | 1 | | 2‡ (RH equivalent to CHC) | 2 |
| | | Nandgaon | | | 1 | 1‡ | 2 |
| | | Sinner | | | | 1‡ | 2 |
| | | Surgana | | | | 2‡ | 2 |
| Total | | | 1 | 3 | 2 | 16 | 39 |

*AFHC numbers during November 2021–June 2022.
†SDH only exist in Maharashtra.
‡RH equivalent to CHC.
AFHC, Adolescent Friendly Health Clinics; CHC, community health centre; PHC, primary health centre; RH, rural hospital; SDH, sub-district hospital.

on the sequence of items, including potential deletion or addition. These inputs were incorporated into the final version of the checklist used for the main study. The two facilities used for the pretesting phase were not included in the main study.

The observations were scheduled in consultation with the concerned authorities (counsellors/ANMs/MOs) at the facility, who were informed of the objective of conducting these assessments. Written informed consent was sought from the concerned staff of the healthcare facility before commencing the data collection. To avoid any potential bias in the responses, the checklist was not shared with the AFHC staff before data collection. The data collection process in each facility lasted for approximately 90–120 min.

## Measures

The RKSK operational framework[6] [13] outlines seven benchmarks designed to ensure the provision of 'friendly' facility-based clinical and counselling services for adolescents at AFHC. Out of the seven benchmarks, only five benchmarks listed in table 2 have been assessed in this study. The assessment of two benchmarks, that is, non-judgemental and competent health service providers, and awareness of the services to the community members

was not feasible. Evaluating these benchmarks would have required exit interviews and observations of interaction between counsellors and adolescents, as well as interviews with community members. Unfortunately, these aspects were beyond the scope of our study, as conducting exit interviews with adolescents required parental consent. However, adolescents often visited the AFHCs either alone or with their peers. Similarly, observing the interaction of health worker and adolescent posed a threat to the confidentiality of the adolescents and could potentially impact their health seeking behaviour. The checklist included for assessing the RKSK benchmarks consisted of 25 items, detailed in table 2 and online supplemental table S1.

Similarly, to assess compliance with international standards, seven out of eight global standards defined in the WHO global standards for quality healthcare services for adolescents[14] were included in the checklist (table 2). Community support (standard 2) was not assessed as exit interviews with adolescent clients were needed. The developed checklist (table 2 and online supplemental table S2) included only 57 items covering the WHO standards relevant to the Indian context.[20]

**Table 2** Variables assessing RKSK benchmarks and international standards

| RKSK benchmarks[6 13] | No of items included* | Objective |
|---|---|---|
| Infrastructure—clean, bright and colourful | 5 | Provision of attractive, clean and comfortable healthcare facility to avail services. |
| Accessibility by adolescents (distance and convenient working hours) | 3 | Ensures easy access to healthcare facility through convenient working days/hours and local transportation. |
| Awareness about the clinic and range of service it provides (Information, Education and Communication-IEC, Proper Signage) | 6 | Provision of clearly visible sign board in regional language and information, education and communication material to provide awareness about the presence of health facility and services available. |
| Privacy and confidentiality | 10 | Ensures privacy in service provision through segregated areas for different services and maintain confidentiality by securing case records and registers under lock and key, accessible to only authorised AFHC staff. |
| Referral from the periphery/community and further referral linkages with the higher facilities and specialty clinics | 1 | Provision of referral services to higher facilities to improve quality of healthcare among adolescents. |
| **Total items** | **25** | |
| WHO global standards for quality healthcare services for adolescents[14] | | |
| Standard 1—Adolescents' health literacy | 3 | To make adolescents knowledgeable about their own health and aware about where and when to receive health services. |
| Standard 2—Community support† | 0 | To ensure that parents, guardians, other community members and community organisations recognise the value of providing health services to adolescents and support all adolescents to use the health services they need. |
| Standard 3—Appropriate package of services | 2 | Provision of package of services including information, counselling, diagnostic, treatment and care at the health facility to fulfil the needs of all adolescents. |
| Standard 4—Providers' competencies | 10 | Healthcare providers' attitudes, knowledge and skills are at the core of quality service provision. This standard ensures that healthcare providers have the technical competence required to provide effective health services to adolescents. Adolescents' rights to information, privacy, confidentiality, non-discrimination, non-judgemental attitude and respect are protected and fulfilled by the healthcare providers. |
| Standard 5—Facility characteristics | 34 | The health facility has convenient operating hours, a welcoming and clean environment and maintains privacy and confidentiality. It has the equipment, medicines, supplies and technology needed to ensure effective service provision to adolescents. |
| Standard 6—Equity and non-discrimination | 4 | Ensures provision of equitable care to all adolescents irrespective of their ability to pay, age, sex, marital status, education, ethnic origin, sexual orientation or other characteristics. |
| Standard 7—Data and quality improvement | 3 | The standard, stresses on the importance of the facility's actions to collect, analyse and use data on cause-specific service utilisation and quality of care, disaggregated by age and sex to support quality improvement |
| Standard 8—Adolescents' participation | 1 | Ensures adolescents' participation in the planning, monitoring and evaluation of health services and in decisions regarding their own care. It also emphasises on adolescents' participation in certain aspects of service provision. |
| Total items | 57 | |

*Details of the items are present in online supplemental table S1.
†Standard 2 not assessed since exit interviews with adolescents were not conducted.
‡
AFHCs, Adolescent Friendly Health Clinics; RKSK, Rashtriya Kishor Swasthya Karyakram.

## Data analysis

Data were entered into EpiInfo[21] and extracted into Excel for coding, cleaning and analysis. For assessing compliance with international standards, a score was created for each standard as described in the WHO scoring guidelines.[22] Each item in the checklist was dichotomous (yes/no) with 'yes' awarded 1 point, and 'no' awarded 0 . A summation of all the questions in each standard was used to obtain the total score for that standard. The maximum score was obtained by adding the number of items in that standard. The percentage score (%) for each standard was calculated by taking the score, scored by the AFHC, dividing it by the maximum score for the standard, and then multiplying by 100 for the standard. Percentage scores (%) were classified as: (1) ≤10%: not meeting standard, (2) 10%–40%: needs major improvement, (3) 40%–80%: needs some improvement and (4) ≥80% or more: meets the standard.[20] Since, no scoring has been developed for assessing compliance to the RKSK benchmarks, WHO scoring guidelines[22] were adopted for assessing compliance of AFHCs with RKSK benchmarks.

## RESULTS

Out of the total 14 clinics included, 8 were from Maharashtra (Nashik: 4 and Yavatmal: 4), while 6 were from Madhya Pradesh (Panna: 3 and Damoh: 3). Two clinics from Maharashtra were additional, due to the presence of AFHCs at additional level of the health system, that is, SDH. Table 3 describes the characteristics of the 14 clinics including functional AFHC days and their operating hours. The maximum duration during which adolescents could avail services in a day was 7 hours, while the minimum duration was 4 hours.

The following section elucidates the compliance of the AFHCs where the assessment was undertaken with national RKSK benchmarks and WHO global standards of quality health services for adolescents.

## RKSK benchmarks

### Infrastructure: clean, bright and colourful

Out of 14 AFHCs, half of the AFHCs required only some improvement (percentage score 40%–80%) in providing clean, bright and colourful infrastructure, while 5 AFHCs (35.7%) met this benchmark as their score was >80%. One AFHC (7.1%) required major improvements as the score was between 10% and 40% and the other (7.1%) lacked compliance as the percentage score was 0 (table 4). The item-wise analysis (online supplemental table S1) spotlights that as per the observations, majority of the clinics (n=11, 78.5%) had clean surroundings and more than half had adequate lighting in the waiting area (n=9, 64.3%) and clean waiting areas (n=8, 57.1%). A significant attention is required regarding toilet cleanliness (n=9, 64.3%), wall paints and furniture (n=8, 57.1%).

### Accessibility by the adolescents (distance and convenient working hours)

Table 4 shows majority of the AFHCs (n=10, 71.4%) had a percentage score >80% thereby meeting the benchmark of being accessible concerning distance and working hours to adolescents. As observed, all the 14 AFHCs were located at a place that was easily accessible via local transport such as buses and auto-rickshaws. Based on observations and interactions with AFHC staff, it showed that in

**Table 3** Descriptive characteristics of AFHCs (n=14)

| Clinic ID | AFHC location at health system | State | No of operational days in week | Total operational hours |
|---|---|---|---|---|
| 1 | Government medical college | Maharashtra | 2 | 6 |
| 2 | District hospital | Maharashtra | 5 | 7 |
| 3 | District hospital | Madhya Pradesh | 6 | 7 |
| 4 | District hospital | Madhya Pradesh | 6 | 7 |
| 5 | Subdistrict hospital | Maharashtra | 6 | 7 |
| 6 | Subdistrict hospital | Maharashtra | 6 | 4 |
| 7 | Rural hospital | Maharashtra | 5 | 4 |
| 8 | Rural hospital | Maharashtra | 6 | 2 |
| 9 | Community health centre | Madhya Pradesh | 6 | 7 |
| 10 | Community health centre | Madhya Pradesh | 6 | 7 |
| 11 | Primary health centre | Maharashtra | 1 | 4 |
| 12 | Primary health centre | Maharashtra | 1 | 4 |
| 13 | Primary health centre | Madhya Pradesh | 1 | 4 |
| 14 | Primary health centre | Madhya Pradesh | 1 | 4 |

AFHCs, Adolescent Friendly Health Clinics.

**Table 4** Performance of AFHCs as per RKSK benchmarks

| | Score percentage | | | |
|---|---|---|---|---|
| | ≤10% (not meeting the benchmark) | 10%–40% (needs major improvement) | 40%–80% (need some improvement) | ≥80% (met the benchmark) |
| **RKSK benchmarks** | **N (%)** | | | |
| Infrastructure—clean, bright and colourful (max. score 5) | 1 (7.1) | 1 (7.1) | 7 (50) | 5 (35.7) |
| Accessibility by the adolescents (distance and convenient working hours) (max. score 3) | 0 (0) | 2 (14.3) | 2 (14.3) | 10 (71.4) |
| Awareness about the clinic and range of service it provides (Information Education Communication, Proper Signage) (max. score 6) | 1 (7.1) | 5 (35.7) | 5 (35.7) | 3 (21.3) |
| Privacy and confidentiality (max. score 10) | 0 (0) | 0 (0) | 13 (92.9) | 1 (7.1) |
| Referral from the periphery/community and further referral linkages with the higher facilities and specialty clinics (max. score 1) | 13 (92.9) | 0 (0) | 0 (0) | 1 (7.1) |

AFHCs, Adolescent Friendly Health Clinics; RKSK, Rashtriya Kishor Swasthya Karykaram.

the majority of AFHCs (n=11) the timing and functional days were as per the RKSK operational guidelines (online supplemental table S1). None of the AFHCs showed non-compliance (percentage score <10%) in terms of accessibility (table 4).

### Awareness and range of services offered (IEC, Proper Signage) at AFHCs

Regarding generating awareness about the clinic using signage and the IEC materials, it was seen that three AFHCs (21.3%) achieved a perfect score of 100% while only one AFHC failed to meet the benchmark, with a percentage score below 10%. Of the remaining AFHCs, 5 (35.7%) needed major improvement as the percentage score was between 10% and 40%, and the other 5 AFHCs (35.7%) required some improvement, having a percentage score between 40% and 80% (table 4). Item-wise analysis (online supplemental table S1) revealed that a significant number of AFHCs (n=10; 71.4%) had external signboards designating them as AFHC. Visibility and inclusion of operating hours were observed in eight clinics (57.1%). The item which lacked in more than half of the AFHCs was the availability of IEC material in regional language (n=9, 64.3%) and education material in the waiting area (n=8; 57.1%)

### Privacy and confidentiality

Overall, the majority of the AFHCs (n=13, 92.9%) required only some improvements in maintaining the privacy and confidentiality of adolescents as the percentage score was between 40% and 80%. Only a single AFHC managed to fulfil this benchmark, achieving a percentage score >80% (table 4). The item-wise analysis (online supplemental table S1) showed that the majority of clinics (n=10, 71.4%) were physically separated from the general Out patient

department (OPD) within the healthcare facility. Almost all AFHCs (n=13, 92.9%) maintained adolescent case records in a secure place, accessible only to authorised personnel, that is, counsellor or MO. This was observed and verified through interactions. Additionally, in 78.5% of AFHCs (n=11), these records and registers were kept under lock and key either by the counsellor or MO.

Observations corroborated by interactions also revealed that during the registration process, the confidentiality of adolescents' identity was maintained. Additionally, no one could see the adolescent client at the time of counselling due to a separate designated area (n=9, 64.3%). However, an area of concern pertains to the provision of separate areas for clinical services (n=10, 71.4%) and the disbursement of commodities (n=9; 64.3%) within the AFHCs (online supplemental table S1).

### Referral from the periphery/community and further referral linkages with the higher facilities and specialty clinics

Only one AFHC attained a perfect percentage score of 100% signifying the successful accomplishment of the benchmark. While, the remaining AFHCs (92.9%) did not meet the benchmark, as ascertained through observations and verified by interactions. The item-wise analysis showed that only one AFHC had the necessary referral guidelines visibly displayed.

### WHO global standard of quality health services for adolescents
*Standard 1: adolescents' health literacy*

Only three AFHCs (21.4%) met this standard with 100% compliance (table 5), Conversely, an equal number of AFHCs (n=3, 21.4%) lacked compliance, achieving a percentage score lower than 10%. The remaining AFHCs required major (n=3, 21.4%) or some modifications (n=5, 35.7%) as their score fall between 10%–40% and

**Table 5** AFHC performance as per the adapted WHO global standards for quality healthcare services for adolescents[14]

| WHO standards | Percentage score | | | |
| | N (%) | | | |
| | ≤10% (not meeting the standard) | 10%–40% (needs major improvement) | 40%–80% (needs some improvement) | ≥80% (met the standard) |
| --- | --- | --- | --- | --- |
| Standard 1—Adolescents' health literacy (max. score 3) | 3 (21.4) | 3 (21.4) | 5 (35.7) | 3 (21.4) |
| Standard 3—Appropriate package of services (max. score 2) | 13 (92.9) | 0 (0) | 1 (7.1) | 0 (0) |
| Standard 4—Providers' competencies (max score 10) | 10 (71.4) | 4 (28.6) | 0 (0) | 0 (0) |
| Standard 5—Facility characteristics (max. score 34) | 0 (0) | 12 (85.7) | 2 (14.3) | 0 (0) |
| Standard 6—Equity and non-discrimination (max. score 4) | 13 (92.9) | 1 (7.1) | 0 (0) | 0 (0) |
| Standard 7—Data and quality improvement (max score 3) | 0 (0) | 2 (14.3) | 10 (71.4) | 2 (14.3) |
| Standard 8—Adolescents' participation (max score 1) | 14 (100) | 0 (0) | 0 (0) | 0 (0) |

AFHC, Adolescent Friendly Health Clinic.

40%–80%, respectively (table 5). The item-wise analysis highlighted in online supplemental table S2 that while a majority of AFHCs (N=8; 57.1%) displayed visible sign-boards indicating their operating hours, more than half of AFHCs (n=8; 57.1%) lacked resources specifically targeting adolescents for information, education and communication in their waiting areas.

### Standard 3: appropriate package of services
Only one AFHC (7.1%) achieved a percentage score of 50%, indicating the need for some improvement due to the score falling between 40% and 80% (table 5). Conversely, the remaining AFHCs (n=13, 92.9%) fell short of meeting the standards, given their percentage scores were less than 10%. The item-wise analysis (online supplemental table S2) highlighted that both reference guidelines (n=13, 92.9%) and SOPs (n=14, 100%) for which services to be offered within the facility and community were not visibly displayed. This was corroborated through interactions.

### Standard 4: providers' competencies
Four AFHCs (28.6%) achieved percentage scores ranged from 10% to 40%, while the remaining ten AFHCs (71.4%) fell short of meeting the standard, receiving percentage scores below 10% (table 5). The item-wise analysis (online supplemental table S2) revealed that low scoring by AFHCs could be attributed to various factors. The absence of well-defined job descriptions for health-care providers such as counsellors (n=13, 92.9%), ANMs (n=14, 100%), specialists (n=14, 100%), nurses (n=14, 100%) and MOs (n=13, 92.9%). In addition, the lack of policies addressing aspects like confidentiality and privacy (n=14, 100%), the provision of free or affordable services

in all AFHCs (n=14, 100%) and a policy commitment to provide health service to all adolescents without discrim-ination (n=13, 92.9%) can also be contributing factor to the low scores. The availability of BMI growth charts for adolescents as a supportive tool for providing quality clin-ical services to adolescents was noted in only three AFHCs (21.5%).

### Standard 5: facility characteristics
Out of the 14 AFHCs evaluated, 2 (14.3%) demonstrated a need for some improvement, as their percentage scores ranged from 40% and 80%. The remaining AFHCs (n=12, 85.7%) needed significant improvements, given their percentage scores were between 10% and 40% in order to meet the established standard for facility char-acteristics (table 5). Item-wise analysis (online supple-mental table S2) spotlighted the reasons for the lack of compliance. This included lack of drinking water facility in the waiting area, the non-display of SOPs/guidelines pertaining to the privacy and confidentiality of adoles-cents, SOPs outlining staff responsibilities, minimising waiting times, how to provide services to adolescents, with or without an appointment. Several other concerns noted in more than 80% of AFHCs were the absence of provision for safe waste disposal, disposal of sharp objects and the availability of essential equipment like measuring tape, stadiometer, contraceptive pills (oral and emer-gency), pregnancy kits and blood pressure measurement instruments etc.

### Standard 6: equity and non-discrimination
The majority of the AFHCs (n=13, 92.9%) failed to meet the established standard for equity and non-discrimination (table 5). Only one AFHC exhibited a limited degree of

compliance, attaining a percentage score of 25%, which falls within the range of 10%–40%. Online supplemental table S2 shows that a significant number of AFHCs (n=13, 92.9%) did not display a policy commitment to provide health services to all adolescents without any discrimination. Furthermore, other policies, guidelines and procedures related to the provision of free and affordable services, along with equitable services irrespective of sex, marital status and age were lacking across all clinics, as both observed and verified through interactions (online supplemental table S2).

### Standard 7: data and quality improvement

The majority of the AFHCs (n=10, 71.4%) exhibited a need for improvement, as their percentage scores ranged between 40% and 80% while two AFHCs (14.3%) achieved a perfect 100% score (table 5, online supplemental table S2). A detailed item-wise analysis showed that all the observed AFHCs (n=14) maintained an enrolment register, and a majority of AFHCs (n=12, 85.7%) possessed counselling registers. However, the stock register was not kept separate for AFHC (n=12, 85.7%)

### Standard 8: adolescents' participation

The assessment of adolescent participation involved inspection of the display of informed consent guidelines or SOPs at the facilities. Unfortunately, none of the 14 observed AFHCs had any such guidelines or SOPs exhibited, resulting in their failure to adhere to the WHO standard for quality healthcare services for adolescents (table 5).

## DISCUSSION

To the best of our knowledge, this is one of the first study conducted in India that examines the compliance of AFHCs with both national benchmarks and global standards for quality care. Evidence from previous studies conducted in India have primarily evaluated AFHC compliance based on WHO standards, Indian Public Health Standards or checklists from the National Health Mission.[23–26] However, adopting a global standard approach further enables critical evaluation of various facets of the quality of AFHS offered at a global level. It would not only help minimise variations in service quality but also assist in comparing results across different countries.

### RKSK benchmarks

Our study revealed that AFHCs were compliant with the national benchmarks, indicating the government's commitment to holistic adolescent health, by setting up AFHCs under RKSK. Successful compliance with RKSK benchmarks was indicated by a host of reasons, including awareness about the AFHC through signage, convenient working hours, functional days and clean surroundings. Similar findings have been observed in the state of Ahmedabad[24] and West Bengal.[27] Moreover, AFHCs in

both the study states were accessible by local transport, unlike AFHCs in Rajasthan, where ease of accessibility was a major concern.[28] Study results showed that the privacy and confidentiality of the patient were maintained, as all case records were kept in secured place and only authorised personnel (counsellor, ANM and MO) had access to retrieve them. Other requisites pertaining to privacy and confidentiality in adolescent services, like 'the provision of dedicated space for adolescents in clinics' have been one of the pressing issues highlighted in the literature.[29 30] Similar to AFHCs in Rajasthan, Jharkhand and Maharashtra,[31] more than half of the AFHCs included in our study, struggled to provide a separate waiting room and a separate area for clinical services and commodity disbursement for adolescents, while contradicting other regional AFHCs in Chandigarh and Kolkata.[32] The lack of these separate areas can threaten adolescent confidentiality and can act as a barrier in seeking and receiving appropriate medical services. Evidence highlights that adolescents who are assured of confidentiality have a greater willingness to discuss sensitive information such as substance use, mental health and sexual history.[33 34] Recently, the Ministry of Health and Family Welfare (MOHFW) released guidelines to set up a model AFHC in each DH, where all the doors and windows of the clinic must have curtains.[35] To address this gap, the low-cost approach can be scaled up in all AFHCs irrespective of the level of the health system. This involves separating areas for clinical services, counselling and commodity disbursement using curtains.

These AFHCs are seen as crucial 'information hubs' that cater to a diverse range of adolescent health needs. IEC is a powerful tool to bring about social behaviour change and promote a culture of 'health-seeking behaviour' among adolescents, with an emphasis on health promotion and prevention.[36] Therefore, the absence of IEC materials, specifically in the regional language at the health facility, can act as a hurdle in utilisation of the health services by adolescents. Studies across various Indian states including Gujarat (Ahmedabad),[24] Rajasthan[28] replicate the finding of limited IEC material at the AFHCs, while AFHCs in West Bengal[27] provided IEC material on SRH. To make the Indian AFHC compliant to this benchmark, key is to have a combination of no-tech and high-tech resources in AFHCs that may assist in catering to a wider range of literacy levels of adolescents. No-tech can include printed IEC materials such as posters, flipbooks[36] and activity books and high tech can include animated movies, short bites on adolescent health issues in the clinic waiting area as seen in clinics of Sweden[37] and Ghana.[38]

Additional lacunae, highlighted as per the RKSK benchmark in our selected AFHC, included the lack of guidelines on services, such as referral guidelines, further limiting the utilisation of quality services. These findings are consistent with other national[29] and international studies[39 40] where most AFHCs lack guidelines and SOPs. These guidelines are a roadmap to a strong connection between various healthcare systems, ensuring efficient

use of medical services. The introduction of referral guidelines enhances effective service delivery by reducing patient waiting time, overcrowding and improving patient in-and-out flow. It is also anticipated to raise the bar for patient management by appropriate use of investigations, patient counselling and other patient-specific programme services.[41] This emphasis the need for referral guidelines for AFHC in India for creating an organised, efficient and patient-centred healthcare system.

## WHO global standard of quality health services for adolescents

Besides, the aforementioned strengths and limitations of AFHCs, identified using national benchmarks, the study also assessed the compliance of AFHCs to WHO global standards for AFHCs. In addition, to some of the overlapping observations with the RKSK benchmarks, findings related to the WHO standards suggest a needed improvement in the observed AFHCs on clear and defined job descriptions, guidelines and SOPs.

Majority of the AFHCs included in the study lacked clearly defined job descriptions for MOs, counsellors, nurses/ANMs, etc and thus might contribute to ambiguity in the delivery of services. To support this finding, a recent assessment in Bhutan highlighted that more than half of healthcare providers (58%) did not provide SRH-related services to adolescents and 74% said their facilities did not have a guidelines or an SOP to provide equal and standard services to adolescents.[2] Similarly, a study in Ahmedabad (India) reveals shortcomings in the knowledge and technical capabilities of healthcare providers in areas related to health promotion, disease prevention and management for adolescents.[24] This underscores the necessity of well-defined and precise job description for each health worker in AFHCs to ensure their effective functioning and and the delivery of comprehensive services to adolescents. Similarly, in AFHCs observed in two study states, the rights of adolescents to information, including privacy, confidentiality, non-discrimination and non-judgemental attitude were not displayed, hindering the structural quality. Likewise, studies from India[28 31] and across the globe have reported judgemental and discriminatory behaviour,[42 43] and breach of confidentiality[42] by healthcare providers. Strong policies and protocols can enhance the quality of healthcare in India as evident in Ghana[38] where adolescents' positive perception of quality healthcare was influenced by fair, non-discriminatory, respectful, competent and trustworthy healthcare providers. Compliance at the global level, therefore, emphasises the need for organising training programmes such as those conducted in Tanzania[44] to improve the youth friendliness of the AFHCs and to ensure that these practices are consistently upheld protocols, guidelines and SOPs in AFHCs. Displaying written statements can help alleviate fear and confusion about services and adolescent rights. Additionally, it also guarantees that health professionals recall their routine work, thereby ensuring continuity of all services rendered meet quality standards, thereby improving adolescent satisfaction.[45]

Moreover, while most of the recruited AFHCs had clean surroundings and proper signage; the absence of other infrastructural characteristics in the two AFHCs such as clean drinking water, safe storage, appropriate disposal of waste, functioning equipment and medical supplies needed maintenance for effective healthcare delivery. The need for infrastructural and medical supplies for effective and appropriate service delivery and adolescent satisfaction has been substantiated in the assessment of services across countries.[38 46] The insufficient medical supplies and equipment is congruent with facilities in Rajasthan (India)[28] and Uganda,[40] Nigeria[47] while contrast to the AFHC in Dehana district, Ethiopia[39] where availability of facility characteristics was found to be appropriate. Further, to support quality improvement adolescent participation holds prime importance, as healthcare services should be participant-sensitive and reflect their needs.

The study findings also showed that the unavailability of guidelines on informed consent to assess adolescent participation, is similar to facilities in the East and Southern Africa Region including[38] and those in Comoros, Mozambique, Zambia, etc, suggesting advocating for the development of youth-responsive health systems rather than just providing healthcare that is friendly to young people.[48]

Going forward, fulfilling certain aspects of national benchmarks is indicative of the Government of India's commitment to strengthening adolescent health. Nevertheless, there is a need to strengthen the existing AFHC as the lack of access to youth-friendly services paves the way for risky health behaviours among adolescents and exposes them to numerous vulnerabilities.[49] The AFHCs can be strengthened using the approach of community engagement and participatory co-design with youth at the centre to maximise access, uptake and coverage[50 51] by fostering ownership and assessing the impact of this approach in India. Furthermore, there is the potential for future research dedicating to establishing unified criteria for India by amalgamation the RKSK benchmarks with the global standards set by the WHO. This endeavour could use a co-creation approach, actively engaging service providers such as MOs, ANMs and counsellors, as well as beneficiaries, particularly adolescents, in the process. This study provides data on the standard of AFHCs in public health facilities in India. The findings should be interpreted with caution due to the limited geographical scope of the study. The sample included one clinic from each level of the health system in the selected study districts of Maharashtra and Madhya Pradesh and thus may not be generalisable to the entire country. In addition, exclusion of exit interviews with adolescent clients and community members did not permit assessment of all national benchmark and global standards. Inclusion of diverse methods like exit interviews with adolescent clients or employing mystery client methodology would

have provided a comprehensive understanding of the healthcare worker's performance and the quality of care provided.

## CONCLUSION

This study provides a window of opportunity for improvement by identifying few gaps in functional AFHCs. It is essential to address these gaps to create adolescent-friendly health facilities that can contribute to the survival, well-being and transformative growth of adolescents in India. Taking cohesive actions by involving multiple stakeholders and using an approach of community engagement and participatory codesign is crucial to ensure that health services become acceptable, equitable and accessible to all adolescents.

**Acknowledgements** We would like to express our sincere gratitude to all the staff of the Adolescent Friendly Health Clinics in Maharashtra and Madhya Pradesh included in this study. We also thank Ms. Gayatri Nayak, Consultant (Public Health Foundation of India) to facilitate the data collection.

**Contributors** MA, SB and DB conceptualised this study. DB and HM were involved in data collection. DB, SB, HM, SK, AM-J and AM were involved in the analysis and interpretation of the findings. DB, SB, AM and HM drafted the manuscript. MA, AM-J and SD critically reviewed the manuscript. DB is responsible for the overall content as the guarantor.

**Funding** This work was supported by Medical Research Council, UK. Grant number MC_PC_MR/P011446/1.

**Competing interests** None declared.

**Patient and public involvement** Patients and/or the public were not involved in the design, or conduct, or reporting, or dissemination plans of this research.

**Patient consent for publication** Not applicable.

**Ethics approval** This study involves human participants and the ethical clearance for the study was obtained from the Institutional Ethics Committee of the Public Health Foundation of India (Reference # TRC-IEC-342.1/17) along with approvals from the Indian Health Ministry's Screening Committee (2017-2250). Participants gave informed consent to participate in the study before taking part.

**Provenance and peer review** Not commissioned; externally peer reviewed.

**Data availability statement** Data are available on reasonable request. Data are available on reasonable request. As per our Institutional's data sharing policy, prior approval is required from the Research Management Committee (RMC) and the principal investigator (PI) of the study. After approval of request from the committee, deidentified data can only be shared on request.

**ORCID iDs**
Deepika Bahl http://orcid.org/0000-0003-3222-5143
Shalini Bassi http://orcid.org/0000-0001-6348-3335
Amanda Mason-Jones http://orcid.org/0000-0002-4292-3183
Monika Arora http://orcid.org/0000-0001-9987-3933

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
