## [Reviewer comments · BMJ Open]

ARTICLE DETAILS

TITLE (PROVISIONAL)	Compliance of Adolescent-Friendly Health Clinics with National and International Standards: Quantitative Findings from the i-Saathiya Study
AUTHORS	Bahl, Deepika; Bassi, Shalini; Maity, Heeya; Krishnan, Supriya; Dringus, Stefanie; Mason-Jones, Amanda; Malik, Anku; Arora, Monika

VERSION 1 – REVIEW

REVIEWER	Chandra-Mouli, Venkatraman Organisation mondiale de la Sante, SRH
REVIEW RETURNED	08-Nov-2023

GENERAL COMMENTS	Feedback to BMJ Open – Compliance of adolescent friendly health clinics with national and international standards: Qualitative findings from the I-Saathiya Study Overall comments: This paper addresses an important subject. Overall, it does a good job. I have three suggestion. Firstly, a huge limitation is that it does not include an observation of health workers in action. Nor does it include mystery clients. The authors should explain the reasons for this. This is so much more important that many of the other attributes assessed. Secondly, the discussion section needs to be revised and restructured. Thirdly, the references need to be reviewed and checked for appropriateness. Specific comments: Page 2: Line 38: Given that you use the term high compliance with benchmarks, please consider using low or poor compliance in line 47 rather than ‘items that demand attention’. Page 3 Lines 33-35: The absence of observation of clinical practice either by an evaluator or a mystery client, or even an exit interview is a serious limitation of the study. It would be important to say why this was not done. Lines 43-47: It would be useful to distinguish between proven medical interventions e.g., diagnosis and treatment of anemia and prevention of unintended pregnancy through contraceptives,, and
---

	intervention delivery mechanisms. AFHS belong to the latter category. Line 58: Please check and confirm if this paper provides information on cost effectiveness. That is not suggested by the title. Page 4 Line 13: Please consider saying that they are meant to provide the range of services listed. Lines 17-19: Please consider saying that they guidance calls for them to be integrated. Line 40: Reference 12 discusses health check ups in Zimbabwe. Please check if it backs up the statement being made. Line 44: The reference is a global normative document. It does not refer to India. Please check and revise if needed. Page 5 The title of the paper does not refer to a study. Please check and confirm if the reference is appropriate in this context. Page 6 No comments. Page 7 Line 26: Please consider deleting reference 14 in this part of the sentence. It is not appropriate here. Lines 39-42: Please clarify which staff you are referring to here. Page 8 Line 33: Please consider using the term lasted. Pages 9-16 No comments. Page 17 Was there an assessment of the health workers actually providing services ? Page 18 Line 31: Please consider replacing the word with the word needed. Lines 38-44: Please clarify whether you mean non display or non availability. Page 19 Line 29:
--	--

	I am not sure if enrolment is the right word here. Please consider and revise if needed. Page 20 No comments Page 21 Line 10: It is not true that most of the AFHS were compliant with national bench marks – toilet cleanliness, privacy in waiting areas, and weak systems of referral are three key areas in which they fell short. Line 42: Please consider saying ‘in contrast to’ instead of ‘while contradicting other’. Page 22 Line 15: Please consider if reference 39 is appropriate here. It addresses validating materials. Line 31: The same point applies here. Line 41: Please consider revising as follows: ...’AFHS lack guidelines and SOPs’. Page 23 It would help if the discussion separates out the points on guidelines and SOPs, displays of principles and procedures, and training/supporting staff. Also, the text does not clearly distinguish having the above three items and applying them. Page 24: Line 19: The point I made above about using the term contrasting rather than the term contradicting applies here as well. Line 55: Reference 51 is a press release. It is not clear what it refers to. Reference 52 is about a study in Zimbabwe. Is that was is discussed in page 25 ?
--	---

REVIEWER	Awang, Hafizuddin Bin Besut District Health Office
REVIEW RETURNED	19-Nov-2023

GENERAL COMMENTS	Were RKSK benchmarks endorsed by the Indian Ministry of Health? Was there strong agreement between RKSK benchmarks with WHO Global Standards for quality healthcare services? Is it possible to developed a newly harmonized Indian criteria rather than combined version of RKSK with WHO criteria?
--

	The data collection process in each facility persisted approximately 90-120 minutes. When were the interview sessions conducted? During working hour? Or after working hour? If sessions were done during working hour, respondents might have been interrupted by their colleagues or superior officers as they are bound to their duty and obligation during office hour. Where were the interview sessions conducted? Outside of health facilities? If conducted within the working environment, respondents might have reporting bias as their superior officers are around them. Based on the data collection method and results presentation, this study is not appropriate to be regarded as qualitative study, since all results are quantifiable (mostly presented as percentage). It is still a quantitative study, using primary data collection through interview-guided checklist. Need to do minor revision on the study design. If researchers intended to do qualitative study, the data from the interview must be transcribed and analyzed thematically. But what I observed in this study, is purely quantitative study based on health facility audit using specially designed checklist.
--	---

VERSION 1 – AUTHOR RESPONSE

Reviewer Comments 1

Feedback to BMJ Open – Compliance of adolescent friendly health clinics with national and international standards: Qualitative findings from the I-Saathiya Study

Overall comments: This paper addresses an important subject. Overall, it does a good job. I have three suggestion. Firstly, a huge limitation is that it does not include an observation of health workers in action. Nor does it include mystery clients. The authors should explain the reasons for this. This is so much more important that many of the other attributes assessed. Secondly, the discussion section needs to be revised and restructured. Thirdly, the references need to be reviewed and checked for appropriateness.

Thank you for the suggestions.

We acknowledge the concern but observing health workers in action could compromise the privacy and confidentiality of adolescent clients. As a result, we have opted not to include this method in our approach. Similarly, adolescent mystery client is a robust approach. However, adolescents often visits these AFHCs alone or with peers. Therefore, due to the lack of parental consent, we could not interact with them, considering ethical considerations.

Response: The discussion section highlights the results concerning the national and global guidelines and lacunae in each have been discussed separately from the perspective of aligning with result structure. Some of the global items are overlapping with respect to guidelines and policies, discussing them independently would lead to repetition and monotony in the text. Therefore, these have been discussed overall. While, within the existing structure we have modified the discussion section for enhancing its comprehension by enumerating the importance of display of guidelines/SOPs for

various aspects of adolescent friendly services.

Thank you for pointing out, the references have been checked and revised for appropriateness.

Comment 1 (Line 38)- Given that you use the term high compliance with benchmarks, please consider using low or poor compliance in line 47 rather than 'items that demand attention'.

Response: Thank you for the suggestion, change has been made in the abstract

Comment 2 (Page 3,Lines 33-35): The absence of observation of clinical practice either by an evaluator or a mystery client, or even an exit interview is a serious limitation of the study. It would be important to say why this was not done.

Response: We agree that this is a serious limitation but due to ethical considerations this not under the purview of the study. Same has been added in the manuscript.

Comment 3(Lines 43-47): It would be useful to distinguish between proven medical interventions e.g., diagnosis and treatment of anemia and prevention of unintended pregnancy through contraceptives,, and intervention delivery mechanisms. AFHS belong to the latter category.

Response: Thank you for suggestion, information has been added

Comment 4 (Line 58): Please check and confirm if this paper provides information on cost effectiveness. That is not suggested by the title.

Response: The reference has been checked and modified for better appropriateness.

Comment 5 (Page 4 Line 13: Please consider saying that they are meant to provide the range of services listed.

Response: The suggested change has been made.

Comment 6 (Lines 17-19): Please consider saying that they guidance calls for them to be integrated.

Response: The suggested change has been made.

Comment 7 (Line 40): Reference 12 discusses health check ups in Zimbabwe. Please check if it backs up the statement being made.

Response: The reference has been changed for better appropriateness.

Comment 8 (Line 44): The reference is a global normative document. It does not refer to India. Please check and revise if needed.

Response: The reference has been changed for better appropriateness. Reference with respect to Indian context has been added.

Comment 9 (Page 5) : The title of the paper does not refer to a study. Please check and confirm if the reference is appropriate in this context.

Response: The line number is not mentioned. However, the references on page 5 i.e. reference no 15,16 and 17 have been checked for appropriateness and all are appropriate to the context

Page 6

No comments.

Comment 10 (Page 7 Line 26) : Please consider deleting reference 14 in this part of the sentence. It is not appropriate here.

Response: The reference has been removed.

Comment 11 (Lines 39-42): Please clarify which staff you are referring to here.
Response: Change has been made for better clarity.

Comment 12 (Page 8 Line 33): Please consider using the term lasted.
Response: The suggested change has been done.

Pages 9-16
No comments.

Comment 13 (Page 17) :Was there an assessment of the health workers actually providing services ?
Response: As mentioned in the manuscript, page 9, these aspects were beyond the scope of our study (reason explained in preceding responses) and hence, there was no assessment of the health workers providing services. There was only an interaction with the health worker depending on their convenience, to verify certain items that could be securely stored and concealed from direct observation. The same is mentioned on page 7 and 8 of the manuscript. The list of items is given in supplementary table S1 that were verified by interactions.

Comment 14 (Page 18 Line 31): Please consider replacing the word with the word needed.
Response: The suggested change has been made.

Comment 15 (Lines 38-44): Please clarify whether you mean non-display or non availability.
Response: The SOPs/guidelines were not displayed as observed by the study team.

Comment 16 (Page 19, Line 29): I am not sure if enrolment is the right word here. Please consider and revise if needed.
Response: Thank you for your suggestion. The term 'enrollment register' is as per the WHO guideline and has been specified in the RKSK operational framework. Hence, the same has been used in the manuscript and for our assessment checklist.

Page 20
No comments

Comment 17 (Page 21)Line 10: It is not true that most of the AFHS were compliant with national bench marks – toilet cleanliness, privacy in waiting areas, and weak systems of referral are three key areas in which they fell short.
Response: Thank you for your suggestion.
The statement was made based on the overall compliance of all the items under the benchmarks. We agree with your suggestion that the above-mentioned areas hold importance, where major improvement is required. However, there were other areas where AFHCs were compliant. The compliance reasons have also been mentioned on page 21. Both the presence and absence of the items under the benchmarks have been mentioned in the discussion section.

Comment 18 (Line 42): Please consider saying 'in contrast to' instead of 'while contradicting other'.
Response: The suggested change has been made.

Comment 19 (Page 22,Line 15): Please consider if reference 39 is appropriate here. It addresses validating materials.
Response: The reference has been rechecked and was found to be appropriate. Please refer to page 2 of the reference.

Comment 20 (Line 31): The same point applies here.

Response: The reference has been rechecked and was found to be appropriate.

Comment 21 (Line 41): Please consider revising as follows: ...'AFHS lack guidelines and SOPs'.

Response: The suggested change has been made.

Comment 22 (Page 23) :It would help if the discussion separates out the points on guidelines and SOPs, displays of principles and procedures, and training/supporting staff. Also, the text does not clearly distinguish having the above three items and applying them.

Response: The discussion section highlights the results for the national and global guidelines and lacunae in each has been discussed separately from the perspective of aligning with the result structure. Some of the global items are overlapping with respect to guidelines and policies, discussing them independently would lead to repetition and monotony in the text. Therefore, these have been discussed overall. While, within the existing structure we have modified the discussion section to enhance its comprehension by enumerating the importance of display of guidelines/SOPs for various aspects of adolescent friendly services.

Comment 23 (Page 24 Line 19): The point I made above about using the term contrasting rather than the term contradicting applies here as well.

Response: The suggested change has been made.

Comment 24 (Line 55): Reference 51 is a press release. It is not clear what it refers to.

Response: The reference has been changed for better clarity

Comment 25 (Reference 52) is about a study in Zimbabwe. Is that was is discussed in page 25 ?

Response: Reference 52 emphasizes the youth engagement in designing interventions to increase access, uptake and coverage of services. The text in the manuscript also proposes the same. Hence, it is supported by reference 52.

Reviewer: 2

Dr. Hafizuddin Bin Awang, Besut District Health Office

Comments to the Author:

Comment 1: Were RSKS benchmarks endorsed by the Indian Ministry of Health? Was there strong agreement between RSKS benchmarks with WHO Global Standards for quality healthcare services? Is it possible to developed a newly harmonized Indian criteria rather than combined version of RSKS with WHO criteria?

Response: RSKS benchmarks have been formulated by the Ministry of Health, Government of India. Refer to the reference "Ministry of Health & Family Welfare, Government of India. Adolescent Friendly Health Clinics (AFHCs): National Health Mission. Available:

<https://nhm.gov.in/index1.php?lang=1&level=3&sublinkid=1247&lid=421>. With respect to the agreement, there were overlap items in RSKS Benchmarks and WHO standards, although WHO global standards were more detailed. We agree with your suggestion that in future research a newly harmonized Indian criteria can be developed adopting a co-creation approach. We have added this in the discussion section.

Comment 2: The data collection process in each facility persisted approximately 90-120 minutes.

When were the interview sessions conducted? During working hour? Or after working hour? If sessions were done during working hour, respondents might have been interrupted by their colleagues or superior officers as they are bound to their duty and obligation during office hour. Where were the interview sessions conducted? Outside of health facilities? If conducted within the working environment, respondents might have reporting bias as their superior officers are around

them.

Response: Yes data collection persisted for 90-120 minutes which included observations and interaction with the health worker only for the verification of the items which were not visible during the interaction. The same is mentioned on page 7 and 8 of the manuscript. The list of items verified is given in supplementary table S1. The interaction was done after the observation at the convenience of the staff to avoid any disturbance to their schedule. Same has been added under the data collection section. For better clarity, the terminology 'interview' has been changed to 'interaction' in the manuscript.

Comment 3: Based on the data collection method and results presentation, this study is not appropriate to be regarded as qualitative study, since all results are quantifiable (mostly presented as percentage). It is still a quantitative study, using primary data collection through interview-guided checklist. Need to do minor revision on the study design.

If researchers intended to do qualitative study, the data from the interview must be transcribed and analyzed thematically. But what I observed in this study, is purely quantitative study based on health facility audit using specially designed checklist.

Response: Thank you for pointing out this error we agree the approach is quantitative and we have changed accordingly, where applicable.

VERSION 2 – REVIEW

REVIEWER	Awang, Hafizuddin Bin Besut District Health Office
REVIEW RETURNED	12-Jan-2024
GENERAL COMMENTS	Dear author, thank you for your revision. All of my previous comments have been addressed accordingly. The latest revised manuscript is acceptable.